# A distributed time-lapse camera network to track vegetation phenology with high temporal detail and at varying scales

Frans-Jan W. Parmentier[1,2,3], Lennart Nilsen[3], Hans Tømmervik[4], Elisabeth J. Cooper[3]

[1]Center for Biogeochemistry in the Anthropocene, Department of Geosciences, University of Oslo, Oslo, 0315, Norway
[2]Department of Physical Geography and Ecosystem Science, Lund University, Lund, 223 62, Sweden
[3]Department of Arctic and Marine Biology, UiT–The Arctic University of Norway, Tromsø, 9037, Norway
[4]Norwegian Institute of Nature Research (NINA), FRAM—High North Centre for Climate and the Environment, Tromsø, 9296, Norway

*Correspondence to*: Frans-Jan W. Parmentier (frans-jan@thissideofthearctic.org)

**Abstract.** Near-surface remote sensing techniques are essential monitoring tools to provide spatial and temporal resolutions beyond the capabilities of orbital methods. This high level of detail is especially helpful to monitor specific plant communities and to accurately time the phenological stages of vegetation – which satellites can miss by days or weeks in frequently clouded areas such as the Arctic. In this paper, we describe a measurement network that is distributed across varying plant communities in the high Arctic valley of Adventdalen on the Svalbard archipelago, with the aim to monitor vegetation phenology. The network consists of ten racks equipped with sensors that measure NDVI (Normalized Difference Vegetation Index), soil temperature and moisture, as well as time-lapse RGB cameras (a.k.a. Phenocam). Three additional time-lapse cameras are placed on nearby mountains to provide an overview of the valley. We derived the vegetation index GCC (Green Chromatic Channel) from these RGB photos, which has similar applications as NDVI but at a fraction of the cost of NDVI imaging sensors. To create a robust timeseries for GCC, each set of photos was adjusted for unwanted movement of the camera with a stabilizing algorithm that enhances the spatial precision of these measurements. This code is available at https://doi.org/10.5281/zenodo.4554937 (Parmentier, 2021) and can be applied to time series obtained with other time-lapse cameras. This paper presents an overview of the data collection and processing, and an overview of the dataset that is available at https://doi.org/10.21343/kbpq-xb91 (Nilsen et al. 2021). In addition, we provide some examples of how this data can be used to monitor different vegetation communities in the landscape.

## 1 Introduction

Remote sensing techniques from orbital and suborbital platforms have vastly improved our understanding of the world's biomes, especially in hard-to-reach regions such as the Arctic. Satellite data indicate that the Arctic has been greening since the 1990s, which has been attributed to an expansion of shrubs in response to temperature increases (Martin et al., 2017). In recent years, reports indicate that this greening has been slowing, or reducing in some regions, which is possibly connected to plant damage caused by extreme winter events (Phoenix and Bjerke, 2016). Some of the observed changes in greenness may

be connected to earlier snowmelt that extends the snow-free season. However, changes in snowmelt timing may also lead to earlier vascular plant senescence (Semenchuk et al., 2016) and changes in vegetation composition (Cooper et al., 2019). Such ground observations need to be taken into consideration when interpreting data from satellites since it remains challenging to detect changes in plant productivity and shifts in the timing of phenological stages from space (Myers-Smith et al., 2020).

Near-ground observations remain essential to fill spatial and temporal gaps, and to correctly interpret remotely sensed vegetation indices to actual changes in ecosystem functioning and composition (see e.g., Anderson et al., 2016; Westergaard-Nielsen et al., 2017).

Orbital and near-surface observation platforms have varying strengths and weaknesses. Satellites provide much needed

information across the whole of the Arctic, and for long time periods, but they have imperfect temporal coverage. Commonly used vegetation indices such as NDVI (Normalized Difference Vegetation Index) are calculated with spectral bands in the visible and near-infrared regions of the electromagnetic spectrum, which is why clear sky conditions are necessary to collect useful data. The Arctic is one of the cloudiest parts of the planet, and this means that – per location – only a few datapoints may be retrieved during summer, and the peak growing season can be missed by days or even weeks. This is particularly an

issue for high Arctic Svalbard (Karlsen et al., 2018) and it prohibits a precise timing of phenological stages, such as green-up and senescence, while complicating the analysis of interannual changes and long-term trends.

Besides large gaps in temporal data, another common issue with remote sensing products is the coarse spatial resolution. The longest available NDVI timeseries, the GIMMS 3g dataset with data going as far back as 1981(Pinzon and Tucker, 2014), has

a resolution of 8x8 km, composed of an upscaling from the original 1x1 km data collected with the AVHRR sensor (Advanced Very High-Resolution Radiometer). The MODIS (Moderate-resolution Imaging Spectroradiometer), Landsat and Sentinel-2 products have higher resolutions, ranging from thousands to hundreds of square meters, but this level of detail is still not high enough for most arctic landscapes. Arctic ecosystems are highly heterogeneous, particularly in the presence of permafrost, and vegetation composition can vary strongly at the decimetre scale (Davidson et al., 2016). Worldview-2, one of the latest

additions to the DigitalGlobe constellation of satellites, does reach a horizontal resolution of ~30 cm, but revisit times are extremely low and only one high quality image, or less, may be obtained per summer (Bartsch et al., 2016). High resolution imagery at a frequency on par with coarser satellite products has only recently become available, through Planet's Skysat constellation of satellites, but persistent cloud cover remains an obstacle to regular surface monitoring. Satellites are excellent platforms to monitor vegetation consistently over decennia and integrated over large areas, but for the monitoring of specific

plant communities at both high spatial and high temporal resolutions, near-surface observations remain superior.

For example, UAVs equipped with imaging sensors can be used to map vegetation at a field site in high detail – with a spatial resolution of centimeters. Still, they can only be flown under favorable weather conditions and require manual operation, which restricts their use to – often short – field campaigns. It can therefore be advantageous to fix imaging sensors to a mast or

another stationary structure. In that case, equipment can operate autonomously and continuously, does not suffer from data loss due to cloudiness, and can be pointed to specific areas with known species composition. While the footprint of such a setup is relatively small, it delivers information at both high spatial and high temporal detail. Time series measured with near-surface sensors can deliver valuable data that complement and help interpret the large-scale perspective of satellite platforms.

To increase the value of monitoring at the small scale, it is important to cover as many vegetation types as possible within a study area to be able to upscale to a larger, regional context. Unfortunately, high resolution imaging sensors capable of measuring NDVI can be costly and the acquisition of dozens of sensors may not be possible within a typical research budget. However, recent studies have shown that it is possible to calculate vegetation indices with similar applications as NDVI, such as GCC (Green Chromatic Coordinate or Green Chromatic Channel), from photos taken with ordinary RGB cameras,
commonly known as a Phenocam (Anderson et al., 2016; Gillespie et al., 1987; Sonnentag et al., 2012; Westergaard-Nielsen et al., 2017). This makes it possible to deploy a large number of cameras for the fraction of the budget needed to acquire specialized NDVI imaging sensors. A major added benefit of photographs, compared to bulk NDVI measurements, is the capability to track specific plant communities by specifying a region of interest (ROI). Moreover, this method can be used to infer changes in carbon exchange rates (Graham et al., 2006; Wingate et al., 2015) and to differentiate between plant species
(Nagai et al., 2011).

In this paper, we describe a multi-year dataset (2015-2018) of RGB photographs from the high Arctic valley of Adventdalen on Svalbard. Throughout this valley, racks were installed with off-the-shelf RGB time-lapse cameras. For comparison, these racks were complemented with measurements of NDVI, soil temperature and moisture, and thermal infrared. In addition to
these near surface setups, landscape cameras were installed on top of nearby mountains to provide an overview of the valley, and to calculate greenness indices at a landscape scale. This paper specifies how the data was collected and processed, and briefly discusses how these cameras can be used as both a supplement and replacement for satellite data. This dataset will be updated in the future with data from following years (2020 onwards) – according to the protocol laid out in this paper.

## 2 Methods

### 2.1 Site Description

The camera racks were installed across the valley of Adventdalen on the Svalbard archipelago (78.17 ºN, 16.07 º E), as listed in Table 1. The Adventdalen valley is nearly 30 km long and roughly 3 to 4 km wide, the central part of which is dominated by the braided river Adventelva, where vegetation is virtually absent. Up the sides of the valley, slopes become steeper, and vegetation is sparse and scattered due to erosional slope processes. Most vegetation is found in between the river and the steep
sides of the valley, on raised terraces that consist of fluvial and eolian silt (Gilbert et al., 2018) and along shallow stream beds of tributaries to the Adventelva that originate from surrounding valleys. The monitoring experiment focuses on these well-

vegetated parts of the valley, which are large and flat enough to be adequately captured by satellites with a resolution of hundreds of meters or less (i.e. MODIS and higher resolution products). The setup used during the summer of 2018 is depicted in Figure 1.


The vegetation composition in the valley is dominated by three dwarf shrub species (*Salix polaris, Cassiope tetragona* and *Dryas octopetala*), herbs, sedges, rushes and grasses (such as *Eriophorum scheuchzeri*, *Luzula confusa*, *Alopecurus ovatus*, *Dupontia fisheri* and *Poa* spp.). Bryophytes and lichens are common throughout the area. The species distribution differs with surface wetness, which is mostly governed by the microtopography. Raised areas, e.g. on the rims of ice wedges, are generally

well-drained and favourable to dwarf shrubs while depressions are typically wet and dominated by sedges and mosses. A detailed vegetation description for each measurement location is provided in Table 2.

The vegetation types at our measurement locations are relatively common to Svalbard but also the rest of the Arctic. In Table 2 we show that our plots cover 6 (out of 11) vegetation classes defined for Svalbard (Johansen et al. 2012), and correspond to

three classes of the Circumpolar Arctic Vegetation Map (Walker et al. 2005; Raynolds et al. 2019). These are: Sedge/grass, moss wetland vegetation (W1), Graminoid, prostrate dwarf-shrub, forb tundra vegetation (G2), and Prostrate/Hemiprostrate dwarf-shrub tundra (P2). Furthermore, our plots show strong similarities to two more vegetation classes: Rush/grass, forb, cryptogam tundra (G1) and Prostrate dwarf shrub, herb tundra (P1). Combined, these vegetation classes cover nearly a quarter of the unglaciated parts of the Arctic, mostly in Greenland and the Canadian Arctic Archipelago, but also the northernmost

parts of Alaska and Russia. This underscores the relevance of this data to studies of arctic change. In addition, the techniques presented here are applicable to any short stature vegetation type – including grasslands, heaths, croplands and wetlands across the world.

## 2.2 Instrumentation

### 2.2.1 Near-surface racks

The racks on which the instrumentation was installed consisted of sturdy metal poles about 2 m high with two arms extending at the top, oriented at a 90º angle to each other (Figure 2a). Part of the installation was previously described in Anderson et al. (2016) – i.e. the configuration used in 2015. In that year, 5 racks were in use on which GardenWatchCam time-lapse cameras (Model GWC001, Brinno Inc., Taiwan) were installed. These ordinary cameras have a resolution of 1.3 Megapixel (MP), and

RGB-derived indices showed a good correlation with bulk NDVI measurements (Anderson et al., 2016). The cameras took photos at a 4-hour interval and were aimed straight down (i.e., in a nadir orientation).

In 2016, the setup was extended to a total of 10 racks. On the new racks, numbers 6 to 10, a WingScapes TimeLapseCam (WCT-00122; Ebsco Industries, China) was used. This camera, with a resolution of 8 MP, was installed in the same nadir orientation and took photos every six hours (midnight, 6 am, noon and 6 pm). Because of the higher resolution, and better durability, all racks were reconfigured with the WingScapes camera in 2018, and the GardenWatchCam was discontinued – with the exception of rack 1. Both camera types were used at their highest image quality setting, with default settings that do not include automatic white balancing since this has been pointed out as essential to achieve a consistent sensor response (Richardson et al., 2019). The precise use of the specific type of RGB camera for each year is listed in Table 3.

In addition to the RGB cameras, the racks were equipped with Decagon SRS-NDVI sensors (Decagon Devices, WA, USA), which measure spectral reflectance at 630 nm and 800 nm. The NDVI sensors were placed in a recommended off-nadir position of 18º, at a height of 2 m, and covered a circular area of ground approximately 1.3 m in diameter. Hemispheric sensors measured incoming radiation at the same wavelengths, to calculate reflectance, and these were placed on racks 2, 6 and 10. These measurements were used for nearby racks without a hemispheric sensor, since incoming radiation doesn't vary as much spatially as surface reflectance does.

The racks were also equipped with soil moisture and temperature sensors installed at a depth of 10 cm (5TM; Decagon Devices, WA, USA), and a thermal infrared radiometer (SI-400 series; Apogee Devices, UT, USA) that was installed next to the NDVI sensor, pointing in the same off-nadir direction. All data from the Decagon sensors were recorded at 4-hour intervals on an Em50 logger (Decagon Devices, WA, USA). Table 3 lists, for each rack, which sensors were installed in a particular year.

Most racks were kept in the same location from year to year, but some needed to be relocated. Rack 5 was moved in 2016 to a wet meadow to include a moister vegetation type in the data coverage. Rack 8 was moved in 2017 to a location close to an eddy covariance tower (see Pirk et al., 2017), to be able to compare the measurements to ecosystem carbon fluxes. In 2017, all racks received a new base to forego the need for guywire. To ease installation of this upgrade, some racks were moved a few meters but kept in the same vegetation type. Minor adjustments were made to the position of some racks in 2018 (see also Tables 1 and 2).

**2.2.2 Landscape cameras**

To connect the detailed coverage of the racks to the larger scale, a few landscape cameras were placed on nearby mountains (see Figure 1). Initially only on a mountain called Breinosa, close to racks 1 to 5, but later also on two additional mountains, Bolternosa (pointing to rack 8) and Lindholmhøgda (pointing to rack 6). The camera on Breinosa was operational in all years, and at Bolternosa in 2017 and 2018. The camera at Lindholmhøgda was installed in both 2016 and 2018, but no data was collected in 2018 due to equipment malfunction.

In 2015, a multispectral camera was used on Breinosa (Agricultural Digital Camera, TetraCam Inc., CA, USA) which has a resolution of 3.2 MP (2048x1536 pixels). In this camera, the blue channel had been replaced with a near infrared band (sensitive up to 920 nm), which makes it possible to calculate NDVI. Photos were taken at 11:00, 12:00, 13:00 and 14:00. For better comparison with the near-surface racks, and because of their higher resolution, this camera was replaced in 2016 with the same WingScapes camera used on the racks. The landscape camera on Lindholmhøgda was also a WingScapes. In 2017, this type of camera was placed on both Breinosa and Bolternosa. Photos were taken each day at 6 AM, noon and 6 PM. In 2018, these cameras were upgraded to CuddeBack E2 timelapse cameras (CuddeBack Digital, WI, USA). These cameras have a resolution of 20 MP, which strongly improved the ability to resolve small-scale spatial variations in vegetation composition. No automatic white balancing was used on any of these cameras.

## 2.3 Data processing

### 2.3.1 Pre-processing and stabilization

After data collection, the photos were manually checked to ensure that they were of high quality. Photos were filtered out because of, for example, snow on the ground, water droplets on the lens, or darkness when polar day ended in late summer. In a few instances, photos were removed if the contrast was too high due to bright sunlight. This was mostly necessary at low sun angles when shading can be problematic. For the landscape cameras, the high contrast was also an issue when there were scattered clouds or when the mountains cast long shadows. For these photos, images with snow on the ground were retained to show how snowmelt differs across the landscape. The filtering mostly led to short gaps, typically no more than one or two days. This was acceptable considering the slow change in the vegetation indices.

After this initial screening, the photos needed to be corrected for unwanted movement of the camera to ensure, as much as possible, that each pixel in the photo corresponded to the same area on the ground. This correction was necessary for the first two years in particular, when the racks were held upright with guywire. This guywire was prone to slackening, allowing the racks to move. This led to a shift over time in the surface area observed by the cameras. This issue became particularly problematic in 2016 when the guywire of several racks was completely loosened by reindeer, and the installations rotated away from their initial position. In some cases, the cameras were no longer in a nadir orientation.

Due to these problems, the racks were modified in 2017 and placed on a permanent base without the need for guywire. While this made the racks very stable, some minor displacement was still possible from ground movement related to freeze/thaw processes or slight movement in the orientation of the camera. Similarly, the landscape cameras on top of the mountains were firmly placed on tripods, but some movement, e.g., due to the wind, led to minor shifts in the photos. To compensate for these

unwanted movements, a stabilization algorithm was applied to all photos from all cameras in all years. An example of such a corrected photo is shown in Figure 3.

The algorithm, written in Python, makes use of OpenCV (Open Source Computer Vision Library), an open source computer vision and machine learning software library (Bradski, 2000). OpenCV includes modules for feature tracking and image alignment that can be used to adjust for any yaw, pitch and roll movements and lateral shifts of the cameras and the racks. To find the movement between two successive photos, they were first converted to grayscale and the histogram of both photos was equalized. This minimizes differences between photos due to varying light conditions. Also, a mask was applied to ignore
features of the installation itself, such as the rack and guywire.

Once two successive photos were treated this way, a Harris corner detector algorithm was applied to identify features that could be tracked between both photos (Harris and Stephens, 1988). This could be, for example, a small stone, a crack in the soil or a twig. After the corners of these features were identified in both photos, the optical flow between the two was calculated
with the method described by Lucas and Kanade (1981). The optical flow was used to calculate an affine transformation between the two photos. This kind of transformation is used to rotate an image within three dimensions while preserving straight lines and surfaces. Once the affine transformation was applied, the next photo was imported, and the procedure was repeated.

For the mountain cameras, a slightly different method was used. The feature identification and optical flow calculation that was used for the racks was not applicable, since the algorithm would try to correct the pitch between photos (i.e., the angle between the valley floor and the camera). However, due to the large distance to the mountain and a sturdy installation on a tripod, this angle was fairly constant. Typically, photos would differ in alignment by a few pixels only and small lateral adjustments along the x- and y-axes were sufficient to align the photos. Therefore, an algorithm was applied that originally
was developed to compose HDR photos (Ward, 2012) but excellently suited for our purposes, since it returns a lateral shift in pixels along the x and y axes of a photo and it is insensitive to changing light conditions. In one or two cases, as determined by a visual check, a rotation needed to be manually specified (determined through trial and error) because of slight rolling of the camera. From these x-y shifts and rotation angles, an affine transformation was composed and applied to the mountain photographs.

While the algorithms automated the alignment of the photos, they still needed a thorough check afterwards. Since the affine transforms were applied cumulatively, small mistakes in the alignment could add up to an incorrect result by the end of the summer. Automatic alignment was difficult in plots that lacked strong features to track between photos, e.g., with a lot of moss and grasses, as well as in situations where differing lighting conditions cast shadows that were incorrectly identified as

movement. Sunny days were problematic in particular, but a layer of rime in the morning or a wet soil after rain could also lead to large differences between photos that prevented an automatic adjustment.

To resolve this problem, these photos were either filtered out or it was indicated in the script that an affine transform wasn't necessary at those instances. In some cases, for example the racks that had become loose in 2016, the affine transform had to
be manually specified. This was necessary when the racks had been fixed upright during a field visit, and the shift between photos became too large for the algorithm to process.

Despite the need for the trial and error afterwards, the automatization made it possible to align all plots relatively quickly and spatial differences between the first and last photo were typically no larger than a couple of cm on the ground, and often less.
This made it possible to track individual plants throughout the growing season, but for the purpose of this paper we will use greenness indices determined over as large an area as possible.

The quality check on the data from the NDVI sensor was limited to the removal of spikes in the data. Outliers were determined by analyzing the reflectance for the 630 nm and 800 nm bands separately. Data points that were two standard deviations
removed from the mean, determined across the whole season, were considered outliers and removed. Also, NDVI values were removed if they were negative (typically due to snow cover) or if it was known that snow was present on the ground. Soil and surface temperatures did not show significant outliers, while soil moisture data was only retained for those dates where the soil was unfrozen.

### 2.3.2 Calculation of greenness indices

Previous analysis of the data collected in 2015 showed a high correlation between NDVI and several greenness indices derived from the RGB cameras – i.e. GCC, 2G_RBi and GRVI (Anderson et al., 2016). We determined whether these greenness indices differed between camera types (GardenWatch and WingScapes) by operating these cameras in parallel on rack 1 for a few weeks in 2017. This showed that GRVI differed quite strongly, while GCC was highly consistent between camera types (Figure 4). Since this index also showed lower variance and correlated best with NDVI, when considering all plots, we use GCC
throughout this manuscript. GCC is an index that shows the intensity of the green channel in a photo relative to the sum of the intensities of all channels:

$$GCC = \frac{G_{i,j}}{R_{i,j} + G_{i,j} + B_{i,j}}$$

$R_{i,j}$, $G_{i,j}$ and $B_{i,j}$ are the intensities of the red, green and blue channel at row $i$ and column $j$ of a photograph. GCC was calculated for each pixel in the photograph for as large an area as possible, which was specified with a mask. An example of such a mask

is shown in Figure 3. These masks are not necessarily of the same size/shape in all plots. All masks used to get this data are included in the public data archive for use in further studies. The values obtained from each photo were averaged to find a value for the whole plot.


For the cameras on top of the mountain, the calculation of GCC was the same, and masks could be used to define areas of interest in which species composition is similar. This is useful, for example, to track vegetation communities at a larger scale, and to identify diverging patterns in the landscape. For racks that were placed in an area with rather uniform vegetation, the mountain cameras also open up opportunities to compare patterns directly with the photos taken at the racks.

## 265  3 Dataset overview

Figure 5 shows time series of NDVI and GCC measured at the racks from 2015 to 2018. The patterns of NDVI and GCC show strong similarities, where the timing of the strongest increase and decrease in NDVI corresponds to the strongest change in GCC. At most racks, and in most years, the timing of the peak in NDVI and GCC also corresponds well. The figure also shows that the use of lower resolution GardenWatch cameras on racks 1 to 5, from 2015 to 2017, typically led to more scatter in GCC

than on the setups that used the higher resolution WingScapes camera, but the overall temporal pattern was very similar. The GardenWatch was phased out in 2018 for racks 2 to 5, which is why the scatter in GCC became lower in that year.

In 2017, data collection continued into September and October, a period in which the days are rapidly shortening on Svalbard, and the solar angle is low throughout the day. The low amount of incoming sunlight increases shading, which is reflected in a

larger scatter for both NDVI and GCC. Interestingly, many plots show a hump in NDVI during this time. An early frost period following day 250 (see Figure 6) suppressed NDVI values, rebounding when temperatures rose slightly in the days after. By this time in mid-September, however, vascular plants have already senesced. It became apparent that the slight increase in NDVI may be linked to the changes in air temperature in combination with continued activity by mosses, since they still appeared green in the photos. Nonetheless, a low solar angle leads to a worsening signal-to-noise ratio (Stow et al., 2004),

which is why these late season patterns should be interpreted with care.

While considering spatial and temporal differences, it appears that the relationship between GCC and NDVI is rather consistent from year to year when the same plot is considered. The possible exceptions are plot 2 and 4, which contained a large fraction of bare ground. When the RGB camera and NDVI sensor are not pointing at the exact same area in such heterogeneous

landscapes, the amount of bare ground and vegetation in their field of view will diverge, causing a relative difference in magnitude between the two indices. Moreover, when comparing one plot to another, the relative magnitude of GCC and NDVI is quite dissimilar. This inconsistent spatial relationship between NDVI and GCC is possibly related to different responses in the infrared, which would affect NDVI but not GCC. This may be caused by differences in soil composition, soil moisture or

vegetation composition. Therefore, it is possible that spatial patterns of GCC and NDVI are quite different, despite the fact that their temporal patterns match very well. This suggests that GCC is a useful tool to acquire a more accurate determination of the timing of phenological stages (Brown et al., 2016), but the spatial discrimination of vegetation types and/or biomass based on either GCC or NDVI data may deliver divergent results.

Figure 6 shows the ancillary data collected at the racks since 2016, namely surface temperature, soil temperature and soil moisture. Not all racks were equipped with these sensors. Surface temperature has only been measured at racks 1 to 5 during the project period and, for one season only, at rack 10 in 2016. Unfortunately, the sensors for soil temperature and moisture malfunctioned at most sites in 2016. Racks 6 to 9 had no additional sensors before 2017.

These data show that surface temperature was slightly higher than soil temperature (as expected), and that there was a strong variation in soil moisture among the sites, while for most of them there was little variation during the season itself. One of the few exceptions to that rule was rack 7 in 2018. This rack was placed in a wet vegetation type that had standing water from snowmelt, and this can lead to high soil moisture values at the start of the growing season (Mörsdorf et al., 2019). This early season peak was not captured in the year before, probably because of a late installation of the sensor. In 2018, there was also a peak around day 250, which coincided with rainfall that collected in the area, and some standing water was visible in the automated photographs taken at the rack. Interestingly, this peak in soil moisture did not appear to affect NDVI and GCC to a large degree (See Figure 5). While a small uptick in NDVI is visible, GCC hardly changed at all.

Finally, Figure 7 shows an example of how the mountain cameras can be used to determine landscape-wide changes in GCC by selecting different regions of interest (ROI). The area on the left (outlined in blue) is a dry exposed area with active cryoturbation, leading to patterned ground. As a consequence, vegetation cover is lower than in the rest of the area and this is reflected as a lower value for GCC. Meanwhile, the area outlined in orange is a wet area with productive vegetation, located along a streambed. From the photograph, it is already clear that this area is much greener, and this leads to a higher value for GCC.

These are just two examples of how GCC can be used to track vegetation differences in the landscape. In principle, regions of interest can be drawn otherwise, depending on the purpose. For example, areas that correspond to a pixel from MODIS or Sentinel-2 can be identified to compare directly with satellite data, which helps to set these data in a regional context (see e.g. Hufkens et al., 2012). It must be noted that to scale up from the plot to the landscape scale, and beyond, it's necessary to have a vegetation map to know the differing proportions of each vegetation type. Vegetation types that exhibit similar seasonal patterns in GCC can be grouped through k-means clustering, random forest algorithms and other machine learning tools. This may form the basis for a detailed vegetation map and can be used to track long-term changes in vegetation composition.

While such applications have potential, care needs to be taken when using these indices. The landscape camera takes photos at a low viewing angle, which may lead to different values for GCC than if these photos were taken straight down. Indeed, Figure 7c shows rather high values for GCC – partly due to the productive vegetation – which are higher than at any of the racks. The low viewing angle may obscure bare ground and give a greener appearance to the picture than if viewed directly from above. Also, the use of different cameras (CuddeBack vs WingScapes) may affect GCC differently.

Another issue arises from the differences in the spatial patterns between NDVI and GCC, which may make it challenging to compare vegetation maps that are based on either index. Since NDVI relies not just on the visible part of the electromagnetic spectrum but also the near infrared, it would be expected that differences in the amplitude of these signals arise when responses in the visible and near-infrared bands diverge. However, the main purpose of this dataset is to assess the timing and pattern of phenological stages which are derived from the direction of change in either NDVI or GCC, rather than the absolute magnitude of these indices. Figure 5 shows that the pattern of green-up, peak growing season and senescence compare quite well, as found previously by others (e.g., Richardson et al. 2018; Sonnentag et al. 2012), and we expect that the application of NDVI and GCC to assess phenological timing will be relatively similar across the landscape. Therefore, we consider the two vegetation indices as complementary. While NDVI responds to increased overall growth of vegetation (reflected near-infrared light and absorption of red light), GCC responds to the changing level of green pigments in the vegetation.

**4 Data availability**

The data presented in this paper is publicly available through the online repository of the Arctic Data Centre of the Norwegian Meteorological Institute at https://doi.org/10.21343/kbpq-xb91 (Nilsen et al. 2021) under a CC-BY-SA license. The data collected at each rack, as shown in Figures 5 and 6, is stored in individual NetCDF files that include metadata such as the coordinates, date and time of collection and the instrumentation. The time-lapse photos collected at each rack and from the three landscape cameras are available at the same location – adjusted for rotational and lateral movements – as JPEG images. These images are accompanied by a text file containing all relevant metadata. The masks used to calculate the time series of GCC (shown in Figure 5) are included for the racks, but not for the landscape cameras since regions of interest may differ from user to user and therefore these weren't specified in advance. The photos from the landscape cameras are also available as JPEG images, corrected for lateral movements. The python scripts used to align the photos are hosted on Github and can be downloaded from https://doi.org/10.5281/zenodo.4554937 (Parmentier, 2021) under a standard MIT software license.

**5 Conclusion**

This paper shows how ordinary RGB cameras can be used to identify temporal and spatial patterns in vegetation phenology, through both detailed information at the plot level as well as a broad overview at the landscape scale, and beyond the

capabilities of current satellite products. Similar setups with Phenocams remain scarce in the Arctic, where logistical challenges due to the absence of a reliable power supply and the remoteness of field sites makes the continuous operation of field equipment challenging. Our setup resolves this issue by not only being low-cost but also low maintenance. We further show how unwanted movement by cameras can automatically be compensated for with a stabilization algorithm to achieve consistent imagery and high precision.

The dataset presented here covers the full growing season, with minimal gaps, while satellites may only capture a few datapoints during the same time period due to persistent arctic cloud cover. GCC also compares well to NDVI at the plot level and shows a similar temporal pattern. Still, there are considerable differences in the magnitude of GCC among plots, and its magnitude compared to NDVI equally differs. Care needs to be taken before RGB-derived indices are used to upscale to a larger area, which is why a comparison to vegetation maps, high resolution satellite data and drone imagery should be included in such analyses.

Despite these caveats, the examples presented here show that the ability to collect images at a high temporal resolution and at a low cost, while retrieving scientifically meaningful vegetation indices from specific areas, are major advantages of the use of ordinary RGB cameras (see also Richardson, 2019; Sonnentag et al., 2012). When applied both at the plot level and at the landscape level, as in this study, this relatively low-cost technique has a strong capacity to inform us in detail about changes in vegetation productivity, phenology, and composition, beyond the current capabilities of remote sensing platforms.

**6 Author contribution**

FJWP processed and analyzed the data, developed the code to stabilize the RGB photos, and prepared the manuscript with contributions from all co-authors. LN, HT and EJC designed the experiments, and all authors contributed to the installation and maintenance of the equipment as well as data collection.

**7 Competing interests**

The authors declare that they have no conflict of interest.

**8 Acknowledgements**

This research was funded by the Research Council of Norway (RCN) under project numbers 230970 (SnoEco), 269927 (SIOS-InfraNor) and 287402 (VANWHITE). FJWP received additional funding from the RCN under project number 274711

(WINTERPROOF) as well as the Swedish Research Council under project nr. 2017-05268. The Garden Watch Cameras were kindly provided by Shin Nagai of the Japan Agency for Marine-Earth Science and Technology.

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

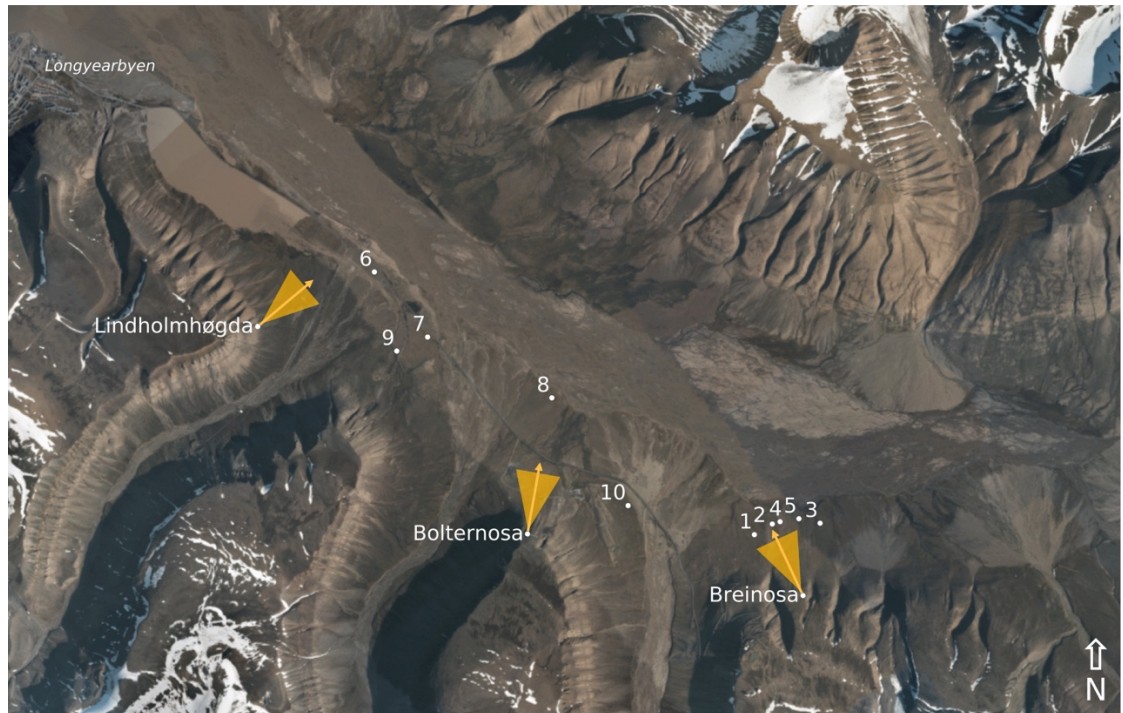

**Figure 1. Locations of the instrumentation in the valley of Adventdalen during the 2018 field season (white dots). The racks are labeled with their respective number while the landscape cameras are labeled according to the nearby mountain peaks. The yellow arrows show the direction in which the landscape cameras are pointed while the triangles indicate their approximate viewing angle. The background image is a composite of several orthographic photographs taken by the Norwegian Polar Institute in July 2009.**

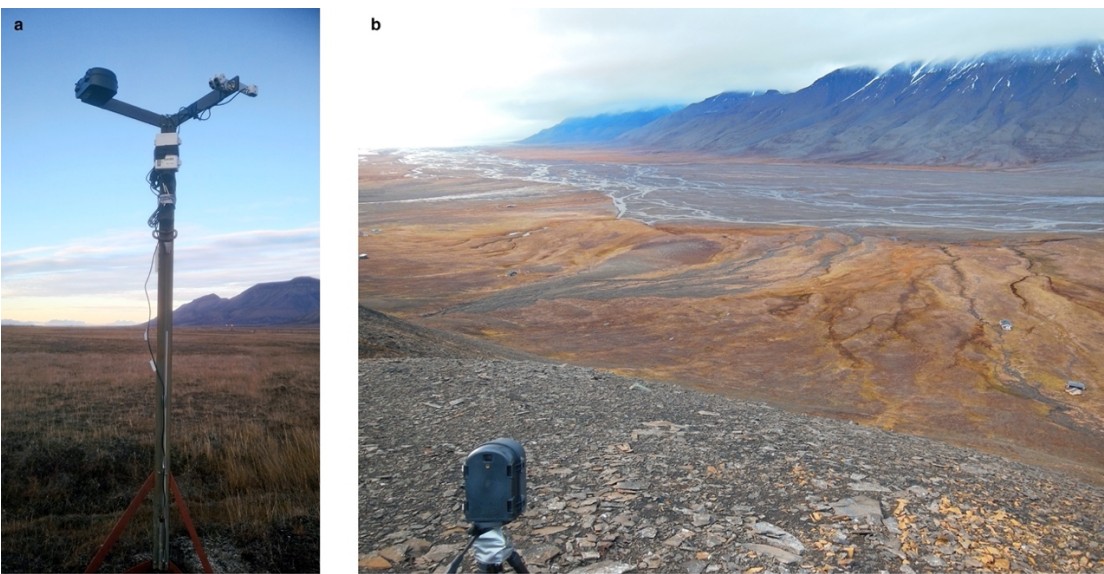

**Figure 2. Overview photographs of a) rack 8 mounted with a WingScapes camera (on the left arm) and Decagon NDVI sensors, both incoming and reflected (on the right arm), and b) a WingScapes camera on top of Breinosa overlooking the area with racks 1 to 5. The picture of the rack was taken in mid-October 2017 and the overview photo of the mountain camera was taken in mid-September 2016. In both photographs, the vegetation had senesced, hence the brown colour.**

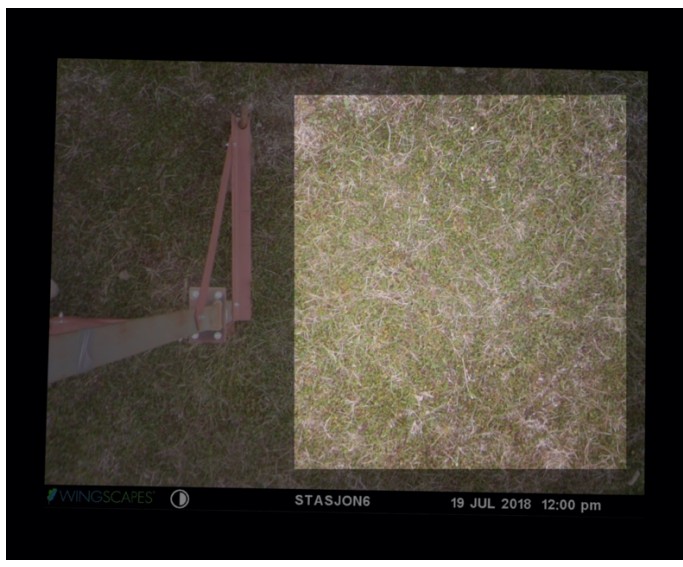

**Figure 3. Example of a stabilized photo for rack 6, taken on the 19th of July 2018 at noon. Slight movements by the thawing of the topsoil turned this camera a few angles out of its original position, which was corrected for by the stabilization algorithm through a rotation of the photo. The bright rectangle indicates the mask for which the greenness index was calculated, which excludes parts of the photo where the rack itself is visible as well as areas where shadows were cast by the rack (darkened regions). The ground surface defined by these masks has been verified to be visible in all photos.**

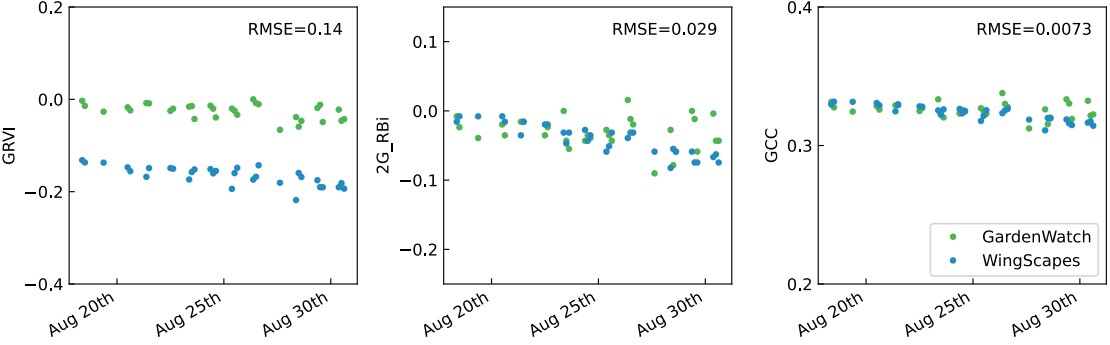

**Figure 4. Comparison of the GardenWatch and WingScapes cameras for three different vegetation indices (GRVI, 2G_RBi and GCC) during a two-week period in August 2017. The root mean squared error (RMSE) between the two cameras is shown in each subplot.**

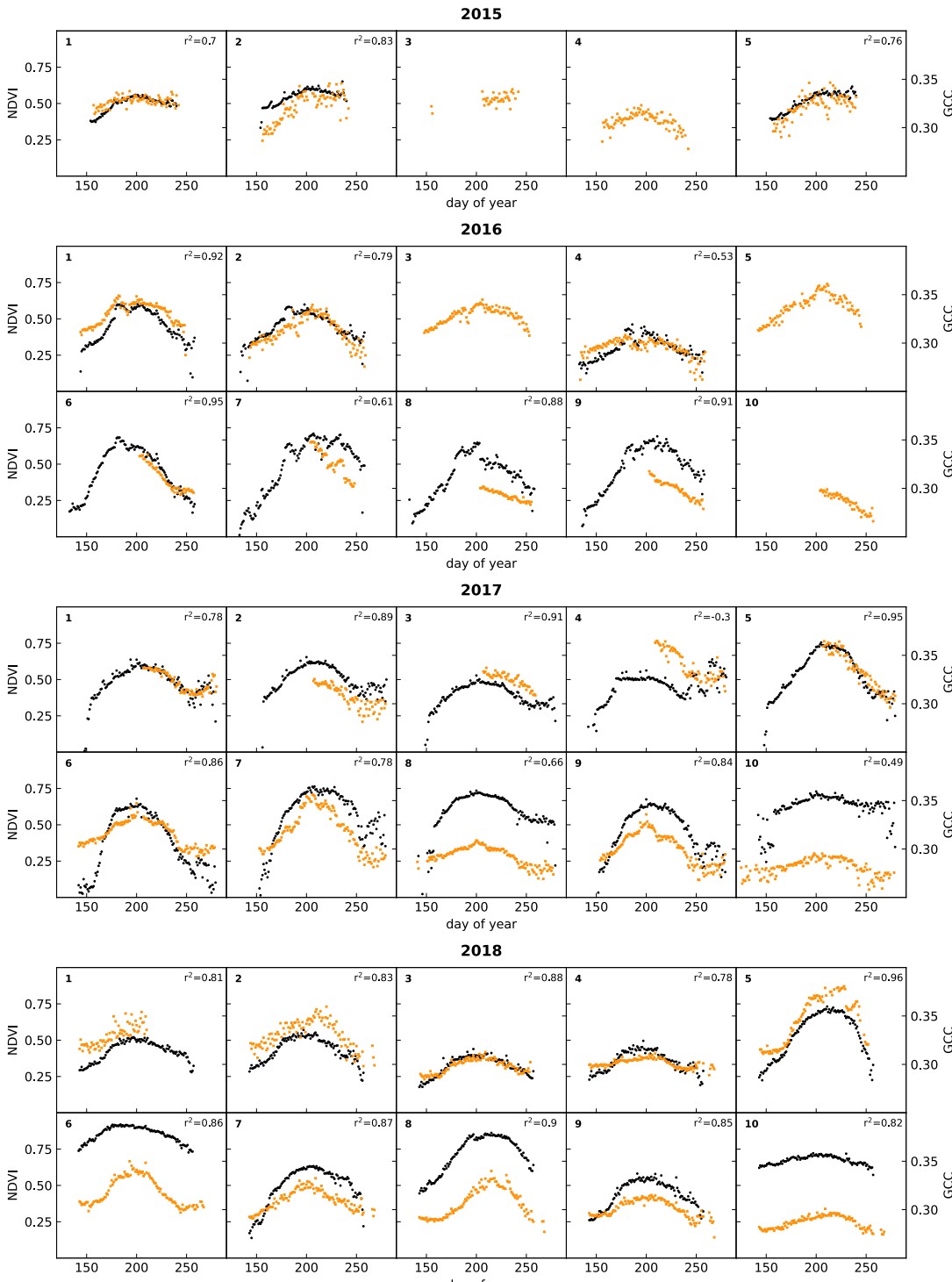

**Figure 5. Time series of daily medians of NDVI (black) and GCC (orange) for all racks (labelled in the top left corner of each subplot). Correlations between NDVI and GCC are also indicated. See Table 3 for details on the type of RGB camera used.**

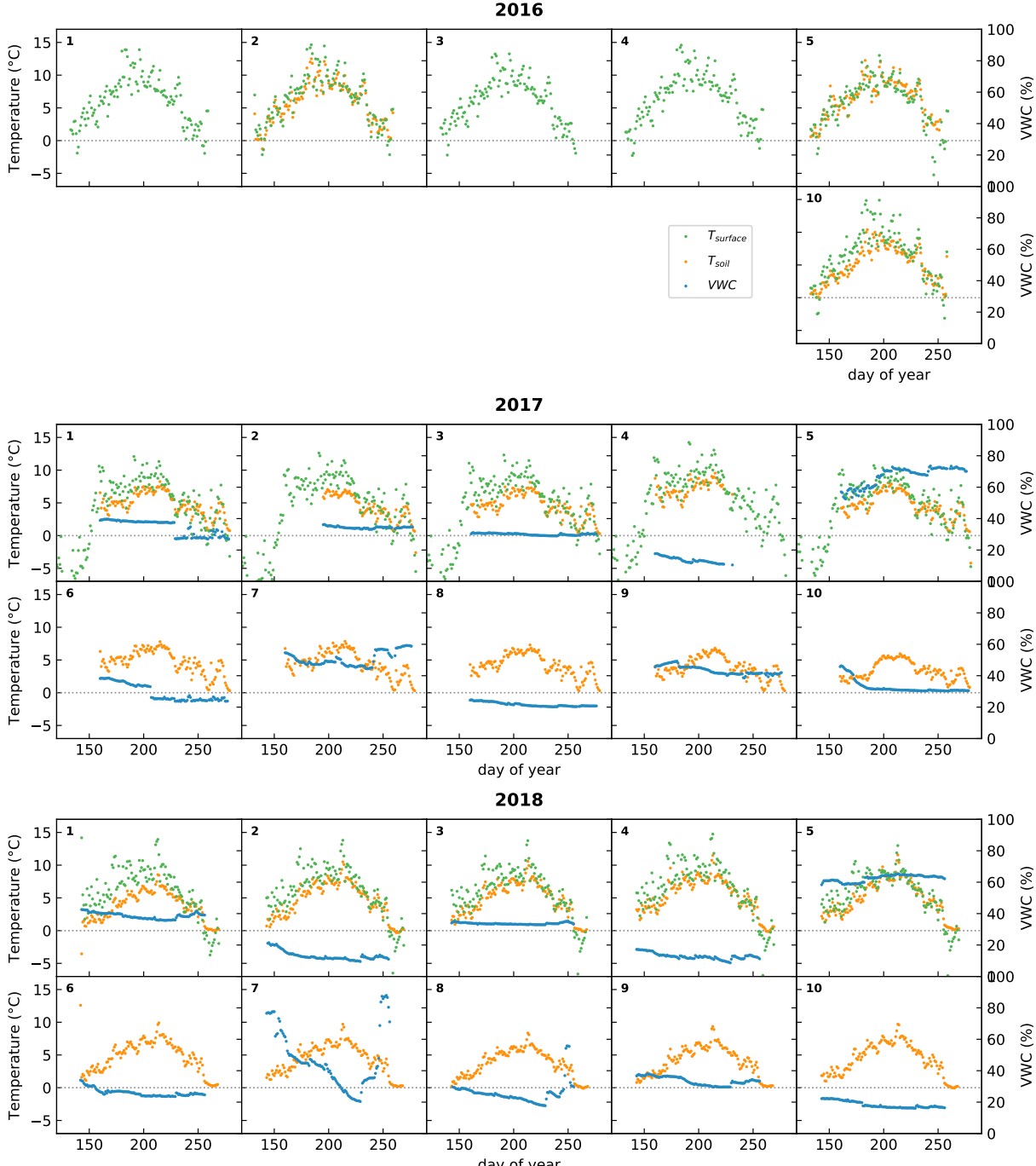

**Figure 6.** Time series of daily averages of surface temperature (green), derived from a thermal IR sensor, and in-situ measured soil temperature (orange) and volumetric water content (VWC, in blue) at a depth of 10 cm. The dotted line indicates 0 °C. In 2016, the soil moisture sensors, and some of the soil temperature sensors, malfunctioned and are not plotted. Rack 6 to 9 had no additional sensors during 2016 and are not shown. The numbers in the top left corners of the subplots indicate the racks on which these sensors were installed.

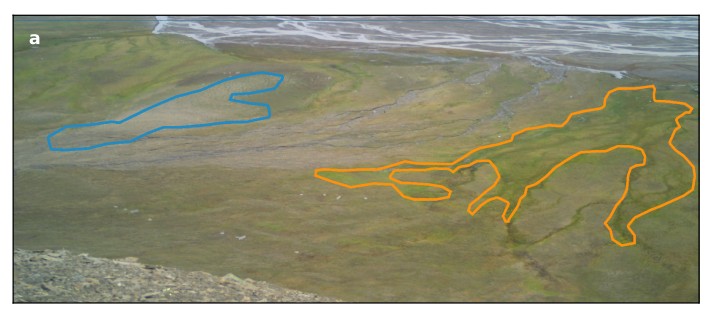

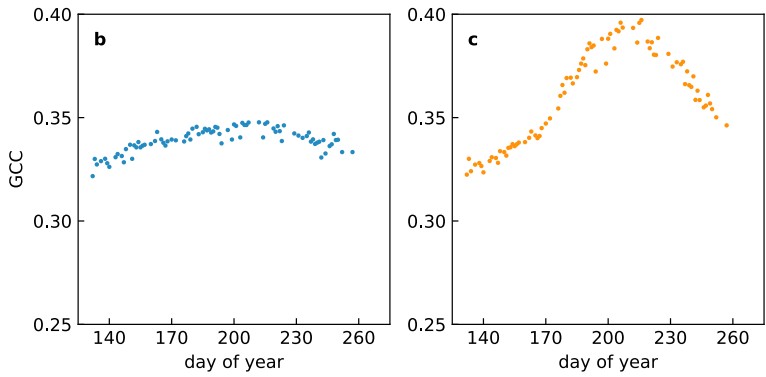

**Figure 7. Examples of RGB indices derived from different regions of interest (ROI) during the 2018 growing season. The blue encircled area (a) has a low vegetation cover, and a relatively high amount of bare ground, which leads to low values of GCC (b). The orange outline encircles dense vegetation growing along a streambed, which leads to higher values for GCC and a more pronounced seasonal pattern (c). The background photograph in (a) was taken on July 19th, 2018 (day of year 200).**

| Rack | 2015 UTM X | 2015 UTM Y | 2016 UTM X | 2016 UTM Y | 2017 UTM X | 2017 UTM Y | 2018 UTM X | 2018 UTM Y |
|------|--------|---------|--------|---------|--------|---------|--------|---------|
| *1* | 523620 | 8677555 | 523620 | 8677555 | 523630 | 8677560 | 523630 | 8677560 |
| *2* | 523854 | 8677689 | 523854 | 8677689 | 523848 | 8677690 | 523848 | 8677690 |
| *3* | 524461 | 8677707 | 524461 | 8677707 | 524451 | 8677710 | 524429 | 8677700 |
| *4* | 523949 | 8677724 | 523949 | 8677724 | 523942 | 8677718 | 523942 | 8677718 |
| *5* | 523943 | 8677249 | 524180 | 8677730 | 524180 | 8677730 | 524172 | 8677754 |
| *6* | | | 519008 | 8680756 | 519008 | 8680756 | 519008 | 8680756 |
| *7* | | | 519655 | 8679964 | 519655 | 8679964 | 519655 | 8679964 |
| *8* | | | 520879 | 8678790 | 521163 | 8679230 | 521167 | 8679224 |
| *9* | | | 519280 | 8679794 | 519280 | 8679794 | 519280 | 8679794 |
| *10* | | | 522013 | 8678008 | 522013 | 8678008 | 522094 | 8677914 |

**Table 1. Location of the camera racks from 2015 to 2018. Two racks were relocated to a different vegetation type (rack 5 in 2016 and rack 8 in 2017). In 2017 and 2018, adjustments were made to the base of the racks and many were moved a few meters within the same vegetation type. A change in the shading indicates that the rack was moved.**

| Rack | Year | Vegetation description | CAVM | SVM |
|------|------|------------------------|------|-----|
| *1* | 2015 - 2018 | Moist moss tundra with *Alopecurus ovatus*, *Bistorta vivipara* and *Salix polaris*. Depressions with *Equisetum arvense*, patches of *Saxifraga hirculus*, and scattered *Dupontia fisheri* and *Eriophorum scheuchzeri*. 100% vegetation cover. | G2 | 12 |
| *2* | 2015 - 2018 | *Cassiope tetragona–Dryas octopetala* heath in a mosaic pattern. 80-100% vegetation cover with regular, small solifluction polygons. Other species present: *Salix polaris*, *Luzula confusa*, *Cerastium arcticum*, *Oxyria digyna* and *Carex rupestris*. | P2 | 14 |
| *3* | 2015 - 2018 | Mosaic of *Dryas octopetala*, *Luzula confusa*, *Poa pratensis alpigena*, *Alopecurus ovatus*, and other graminoids. Lots of *Salix polaris* and *Bistorta vivipara* on moist to wet moss tundra dominated by silty sand. Small landscape feature dominated by soil frost polygon with little vegetation in the center. | G2 | 12 |
| *4* | 2015 - 2018 | *Dryas octopetala–Salix polaris* vegetation on lower part of a gently sloping alluvial fan. Substrate dominated by sandy gravel and stone. Partly exposed with some dominance of lichen. Scattered *Luzula confusa*, *Bistorta vivipara*, *Stellaria longipes* and *Silene uralensis* ssp. *arctica*. Vegetation cover 70-90%. | P2 | 13 |
| *5* | 2015 | *Cassiope tetragona–Dryas octopetala* heath. Composition very similar to rack 2. | P2 | 14 |
| | 2016 - 2018 | Wetland dominated by the grass *Dupontia fisheri* and mosses. Fresh water running through the vegetation. Lots of *Salix polaris* and *Bistorta vivipara*. Scattered *Ranunculus spitsbergense* and *Eriophorum scheuchzeri*. Vegetation cover 100% with a dense bryophyte layer. | W1 | 11 |
| *6* | 2016 - 2018 | Grass dominated sandy sediment plain. *Festuca rubra*, *Poa pratensis alpigena*, and *Alopecurus ovatus*. Thin organic layer, with lots of *Salix polaris* in between the grasses. Vegetation cover 80-100%. | G2 | 16 |
| *7* | 2016 - 2018 | Wetland vegetation on flat silty and sandy substrate, dominated by large polygon soil patterns. *Puccinellia phryganodes*, *Dupontia fisheri* and *Eriophorum scheuchzeri* in the interior part of polygons. *Ranunculus pygmeaus* and bryophytes like *Scorpidium cossonii* and *Scorpidium revolvens* dominate the wettest part in polygon cracks. | W1 | 10,11 |
| *8* | 2016 | *Luzula confusa–Salix polaris* dominated vegetation on a gentle slope with cryoturbation and some bare soil. Sandy gravel with pebbles and stones. Vegetation in typical mosaic. Tufts with *Dryas octopetala* scattered on tussocks and *Cassiope tetragona* in small depressions. Lots of *Luzula confusa*, *Salix polaris*, *Bistorta vivipara* and scattered *Stellaria longipes*. Some depressions dominated by *Equisetum arvense*. | G2 | 16 |
| | 2017 - 2018 | Graminoid dominated vegetation on silty-sandy plain characterized by large scale polygon cryoturbation. The terrain is gently sloping towards the Adventdalen river. Dominant vascular plants are *Dupontia fisheri* and scattered *Eriophorum scheuchzeri*. Vegetation cover generally 100%. | W1 | 11 |
| *9* | 2016 - 2018 | Heath dominated by *Luzula confusa*. Other species present are *Salix polaris*, *Poa pratensis alpigena*, *Carex arcticum* and bryophytes like *Sanionia uncinata* and *Tomentypnum nitens*. Some cryoturbation and silty soil. 70-100% vegetation cover. | G2 | 16 |
| *10* | 2016 - 2018 | Typical *Cassiope tetragona* heath on a north-east facing hillslope. Lots of *Salix polaris* and scattered *Dryas octopetala* and *Stellaria longipes*. Regularly distributed *Luzula confusa* and patches with *Saxifraga hirculus* and *Festuca rubra*. Dominating moss, between mats of *Cassiope tetragona,* is *Sanionia uncinata*. Vegetation cover 90-100%. | P2 | 14 |

**Table 2. Vegetation composition at each rack from 2015 to 2018. Apart from racks 5 and 8, that were moved to different vegetation types, all racks were kept within the same general area with similar vegetation composition. The vegetation classes are according to those defined by the Cirumpolar Arctic Vegetation Map (CAVM; Walker et al., 2005), and the Svalbard Vegetation map (SVM; Johansen et al., 2012).**

530

| Rack | Camera type | | | | NDVI sensor | | | | Thermal IR sensor | | | | Soil temperature and moisture sensor | | | |
|---|---|---|---|---|---|---|---|---|---|---|---|---|---|---|---|---|
| | 2015 | 2016 | 2017 | 2018 | 2015 | 2016 | 2017 | 2018 | 2015 | 2016 | 2017 | 2018 | 2015 | 2016 | 2017 | 2018 |
| 1 | GW | GW | GW WS | GW | x | x | x | x | | x | x | x | | | x | x |
| 2 | GW | GW | GW | WS | x | x | x | x | | x | x | x | | x | x | x |
| 3 | GW | GW | GW | WS | | | x | x | | x | x | x | | | x | x |
| 4 | GW | GW | WS | WS | | x | x | x | | x | x | x | | | x | x |
| 5 | GW | GW | GW | WS | x | | x | x | | x | x | x | | x | x | x |
| 6 | | WS | WS | WS | | x | x | x | | | | | | | x | x |
| 7 | | WS | WS | WS | | x | x | x | | | | | | | x | x |
| 8 | | WS | WS | WS | | x | x | x | | | | | | | x | x |
| 9 | | WS | WS | WS | | x | x | x | | | | | | | x | x |
| 10 | | WS | WS | WS | | | x | x | x | | | | | x | x | x |

**Table 3. Equipment installed at each rack from 2015 to 2018 (GW = GardenWatchCam, WS = WingScapes). Rack 1 switched from a GardenWatchCam to a WingScapes during the 2017 field season, which is why both are indicated. Soil moisture was not recorded in 2016 due to equipment failure.**