# Peer review of "A distributed time-lapse camera network to track vegetation phenology with high temporal detail and at varying scales"

_Earth System Science Data, 2021_

## Author Comment (AC1)

**Response to comments by Referee #1**

We thank the referee for taking the time to review our manuscript and for providing these useful comments. In this document, we respond to the remaining questions and comments and how we addressed them (referee comments in italic). When we refer to line numbers, we mean those in the revised manuscript.

*This manuscript presents a RGB photo dataset collected from a group of near surface Phenocam installed over high Arctic valley. The dataset contains three-years time series of photos with 4-6 hours temporal resolution, and is used as proxy of NDVI indice to monitor vegetation canopies. It concluded that ordinary RGB cameras are a promising tool to identify temporal and spatial patterns in vegetation productivity and composition at a landscape scale, and that GCC compares well to NDVI at the plot level and shows a similar temporal pattern. The dataset is well described , and the methods for data collection and processing are clearly presented. However, some issues need to be handled before the manuscript can be accepted for publication.*

We are glad to hear the overall positive impression the referee had of our paper. We respond to each issue down below:

*Details:*
*1) The case study area, a valley of Adventdalen on the Svalbard archipelagois relatively small and may not well represent large areas in the Arctic. Suggest to add some discussions on how this case study can be somehow scaled up to larger areas or applied in similar landscapes.*

This is a common discussion when it comes to studies on Svalbard: how representative is it? We have therefore added the following text to the manuscript (line 107), and included the relevant vegetation classes in Table 2:

> The vegetation types at our measurement locations are relatively common to Svalbard but also the rest of the Arctic. In Table 2 we show that our plots cover 6 (out of 11) vegetation classes defined for Svalbard (Johansen et al. 2012), and correspond to three classes of the Circumpolar Arctic Vegetation Map (Walker et al. 2005; Raynolds et al. 2019). These are: Sedge/grass, moss wetland vegetation (W1), Graminoid, prostrate dwarf-shrub, forb tundra vegetation (G2), and Prostrate/Hemiprostrate dwarf-shrub tundra (P2). Furthermore, our plots show strong similarities to two more vegetation classes: Rush/grass, forb, cryptogam tundra (G1) and Prostrate dwarf shrub, herb tundra (P1). Combined, these vegetation classes cover nearly a quarter of the unglaciated parts of the Arctic, mostly in Greenland and the Canadian Arctic Archipelago, but also the northernmost parts of Alaska and Russia. This underscores the relevance of this data to studies of arctic change. In addition, the techniques presented here are applicable to any short stature vegetation type – including grasslands, heaths, croplands and wetlands across the world.

*2) GCC is an index composed of RGB bands of digital photo, while NDVI is calculated with near-infrared and red bands. They may have a high correlation in some cases, but we have to make sure such correlation is consistent across space and time. Please add paragraphs to discuss the issue.*

We agree that this may be an issue and we already discuss this in section 3. However, since this setup is used to identify phenological changes, we are mostly interested in the direction of change and less in the absolute value of the remotely sensed signal. The timing of phenological stages compares quite well, and this has previously been concluded by others as well (see e.g., Richardson et al., 2018; Sonnentag et al., 2012). But to further highlight this issue, we have expanded the last paragraph of section 3 which now reads as follows:

> Another issue arises from the differences in the spatial patterns between NDVI and GCC, which may make it challenging to compare vegetation maps that are based on either index. Since NDVI

relies not just on the visible part of the electromagnetic spectrum but also the near infrared, it would be expected that differences in the amplitude of these signals arise when responses in the visible and near-infrared bands diverge. However, the main purpose of this dataset is to assess the timing and pattern of phenological stages which are derived from the direction of change in either NDVI or GCC, rather than the absolute magnitude of these indices. Figure 5 shows that the pattern of green-up, peak growing season and senescence compare quite well, as found previously by others (e.g., Richardson et al. 2018; Sonnentag et al. 2012), and we expect that the application of NDVI and GCC to assess phenological timing will be relatively similar across the landscape. Therefore, we consider the two vegetation indices as complementary. While NDVI responds to increased overall growth of vegetation (reflected near-infrared light and absorption of red light), GCC responds to the changing level of green pigments in the vegetation.

Also, we address the possible issue of misalignment of the NDVI sensor and RGB cameras on line 330:

While considering spatial and temporal differences, it appears that the relationship between GCC and NDVI is rather consistent from year to year when the same plot is considered. The possible exceptions are plot 2 and 4, which contained a large fraction of bare ground. When the RGB camera and NDVI sensor are not pointing at the exact same area in such heterogeneous landscapes, the amount of bare ground and vegetation in their field of view will diverge, causing a relative difference in magnitude between the two indices.

*3) Page 3, line 80-81: Please explain why the dataset ends in 2018, any plan to continue and update?*

The data is still being collected but due to equipment failure no data was collected in 2019. The data from 2020 and onwards will be processed at a later time and added to the dataset. We added the following sentence to the last paragraph of the introduction (line 89):

This dataset will be updated with data from following years (2020 onwards) according to the protocol laid out in this paper.

*4) RGB photos were captured with several different models of digital camera, from GardenWatchCam, WingScapes TimeLapseCam, to Breinosa. Suggest to discuss how these cameras are cross-calibrated and how the photos are normalized to ensure consistency.*

We ran a GardenWatchCam and WingScapes in parallel on rig #1 for two weeks in 2017. This showed that, on average, the two had similar values for GCC but that the GardenWatchCam showed more variation (i.e., noise). This general behavior is mentioned in section 2.3.2. However, just like the potential differences with NDVI, this setup was designed to track the phenology of the vegetation, and we do not expect the timing of those stages to be different with different cameras. This is also an argument made by Richardson et al. (2019), who primarily pointed to automatic white balancing as a potential issue that could affect measurements. Therefore, we have added a new figure showing the comparison (Figure 4) and added the following sentence to section 2.2.1 (line 132):

Both camera types were used at their highest image quality setting, with default settings that do not include automatic white balancing since this has been pointed out as essential to achieve a consistent sensor response (Richardson et al., 2019).

And to the end of section 2.2.2 (line 169):

No automatic white balancing was used on any of these cameras.

And the following to section 2.3.2 (line 246):

We determined whether these greenness indices differed between camera types (GardenWatch and WingScapes) by operating these cameras in parallel on rack 1 for a few weeks in 2017. This showed that GRVI differed quite strongly, while GCC was highly consistent between camera types (Figure 4). Since this index also showed lower variance and correlated best with NDVI, when considering all plots, we use GCC throughout this manuscript.

*5) Page 5, line 145-146: I don't quite understand the sentence "... the blue channel had been replaced with a sensor sensitive to the near infrared, ...". Is it a near-infrared band or just somehow sensitive to near-infrared? Please clarify.*

It's a near-infrared band (sensitive up to 920 nm). This has been changed in the text.

*6) Page 5, line 155-156: I would suggest that we keep the photos that contains snow on nthe ground, as snow is such an important feature in the Arctic and such photos may be useful for research.*

Because of ease of use of this dataset, we don't retain photos with snow for the small rigs, also because it interfered with the stabilization algorithm. However, for the landscape photos we did retain those with snow cover visible, precisely because snow is such an important feature, and this shows how snowmelt differs across the landscape. This is now mentioned in the text (line 177) as follows:

> For these photos, images with snow on the ground were retained to show how snowmelt differs across the landscape.

*7) Page 9, line 263-264: The major difference between NDVI and GCC is contributed by the contrast between near infrared and blue bands. Here you need little in-depth discussion on how they differ and therefore influence the correlation between NDVI and GCC in general.*

See our answer to comment #2.

*8) Page 10, line 311-313: With a data format of JPEG images, I wonder how spatial coordinates are provided in the datasets, in metadata? Please clarify.*

Each timeseries of photos is accompanied with a textfile that contains all metadata. This has now been specified in the text (line 345).

---

## Author Comment (AC2)

**Response to comments by Referee #2**

We thank the reviewer for taking the time to evaluate our manuscript and for providing constructive criticism. We have responded to each comment down below (referee comments in italic, line numbers refer to the revised manuscript):

*For more than a decade, digital cameras have been widely used in Europe, North America, and Japan to monitor vegetation phenology and long-term change, but observational sites in the high Arctic have, to date, been sparse. This paper describes a relatively small network of monitoring sites in the high Arctic valley of Adventdalen on the Svalbard archipelago, which thus helps to fill that gap, although the representativeness of the Svalbard relative to the vast (global) Arctic region is unclear. The monitoring sites described feature a range of sensors including soil temperature and moisture, spectral measurements for NDVI, and visible-wavelength digital cameras. These data are potentially useful and valuable for Arctic researchers worldwide, and may spur further efforts to expand Arctic monitoring across North America and Eurasia.*

We are glad to see that the importance of this work is being acknowledged.

*Details of the installations are somewhat lacking. For example, the camera settings (Auto white balance? Contrast adjustment? Sharpening? JPEG compression) need to be described in greater detail.*

Since we aimed for a low-cost setup, these cameras have a relatively basic function set that don't allow for much adjustment. The cameras were used with their default settings at the highest image quality setting, and no white balancing. This is now indicated in the text. See also our answer to comment #4 by referee 1.

Btw, the documentation of the cameras does not indicate whether JPEG compression was applied but from the quality of the photos we suspect this to be minimal. Contrast adjustment and sharpening are not part of the feature set of these cameras.

*Unlike other PhenoCam-type networks, which have generally adopted a standard camera and configuration, the data set described here includes cameras of a variety of makes and models. Since the make and model of camera undoubtedly has an effect on the image quality, RGB sensitivity, and GCC (both in absolute terms and in its variance) (see Sonnentag et al AFM 2012), it would be helpful to have some assessment of how camera choice impacts the underlying data and data quality (this is mentioned in passing on L302). I suspect that some of the inconsistency in relationships between NDVI and GCC (L260+) could be attributed to differences in cameras and/or camera settings.*

We have expanded on this text (line 329 onwards), also in response to comments by referee 1. We acknowledge that differences may arise, but we found these to be small for GCC when we ran two camera models in parallel in 2017 (GardenWatchCam and Wingscapes). The main purpose of this setup is to get a more precise estimate of the timing of phenological stages, rather than the absolute magnitude of GCC. This timing should not be very different between camera models, as others have also found (Sonnentag et al., 2012).

*The image alignment routine sounds very clever and effective, but there are many other aspects of the processing which appear to require human intervention and assessment. The authors might think about how to automate some of these other quality control steps, so that as the data volumes expand over time, the requirement for human intervention doesn't become burdensome.*

A more automated setup would be welcome but there is always an additional check necessary afterwards to verify that everything went ok. The current algorithm is a trade-off between the amount of time it takes to verify the image alignment and the amount of time it takes to develop a perfect

algorithm with little need for supervision. For just these ten rigs, the photos can be processed in one or two days by an experienced person, which is sufficient for our purposes.

*(And somewhat related to this, given that we are now in 2021, it is surprising that that data set extends only to 2018—are there plans to keep this up-to-date in the future?)*

The monitoring is still ongoing, but due to equipment failure there was no data collected in 2019. The data from 2020 and onwards will be quality checked and stored in the same data repository. See also line 87.

*The authors allude (L290+) to the potential to use the camera data to compare with/evaluate satellite data products – it seems that some analysis of this type could have been done here, as a concrete example, demonstrating proof-of-concept. (Note that in more temperate regions, but in some cases including Arctic sites, many previous papers have conducted this sort of analysis already: Hufkens et al RSE 2012, Klosterman et al Biogeosci 2014, Zhang et al AFM 2018, Richardson et al Sci Reports 2018, and others).*

This is indeed one of the possible use cases of this setup, and a direction for future research. We have cited Hufkens et al. (2012) and Richardson et al. (2018) where relevant.

*The first sentence of the conclusion – "This paper shows that ordinary RGB cameras are a promising tool to identify temporal and spatial patterns in vegetation productivity and composition at a landscape scale" — is misleading. First, one might think that this paper represents the first time that this kind of monitoring has been attempted and proven successful, which is incorrect. Second, there was no quantitative analysis of differences in productivity in relation to differences in camera-derived indices presented here; Figure 6 is entirely anecdotal. I would therefore encourage re-framing the conclusion in a way that accurately represents the novel contributions of this work.*

We have changed the first paragraph of the conclusion to:

> This paper shows how ordinary RGB cameras can be used to identify temporal and spatial patterns in vegetation phenology, through both detailed information at the plot level as well as a broad overview at the landscape scale, and beyond the capabilities of current satellite products. Similar setups with Phenocams remain scarce in the Arctic, where logistical challenges due to the absence of a reliable power supply and the remoteness of field sites makes the continuous operation of field equipment challenging. Our setup resolves this issue by not only being low-cost but also low maintenance. We further show how unwanted movement by cameras can automatically be compensated for with a stabilization algorithm to achieve consistent imagery and high precision.

*More generally, it seems that the seminal literature on using digital imaging to track vegetation on ≈ seasonal time scales could (should?) have been cited in various places throughout the manuscript. For example, Eric Graham's classic "Moss Cam" (Int J Plant Sci 2006) paper would seem highly relevant, even though that analysis was conducted in a more temperate setting. Likewise, with the exception of the paper by Brown et al 2016, the numerous papers that really established the viability of camera-based monitoring of vegetation (and the relation of GCC and other indices to e.g. ecosystem-scale fluxes of carbon and water and satellite-based products) by Nagai, Wingate, Richardson, Morellato and colleagues have been strangely ignored.*

We thank the referee for reminding us about these papers. We have cited Graham, Nagai, Wingate, Richardson and Sonnentag throughout the manuscript when relevant (see also References).

*In terms of the data set itself, I was unable to figure out how to access it (this may be my own incompetence). The landing page (https://adc.met.no/datasets/10.21343/kbpq-xb91) is informative enough, but then the catalog resource*

*(https://thredds.met.no/thredds/catalog/arcticdata/infranor/SnoEco/catalog.html) is cryptic. The data sets are stored as NetCDF files, which is a commonly used format, but not one I regularly use. The Java viewer that can be downloaded won't run under the current Mac OS because Java support has been discontinued. In the end, I wasn't actually able to download a file and see what was in it.  While I understand the attraction of the netCDF format, it also seems that it might be easier for many data end-users if the basic arrays could also be included as flat ASCII or .csv files.*

It is unfortunate that the referee was unable to access the data. We are restricted here by limitations of the online catalog, which only allows for the storage of data in the netCDF format (it actually took quite some effort on their part to include JPEG images as well). We do hope that the website will be updated so the online viewer will also work in internet browsers without the need for Java.

Finally, we want to inform that during a final check of the database we learned that plot 2 and 4 had been switched around for the year 2018 by accident. This will be fixed in the online database. The figures in the paper have been updated accordingly.